# Maternally Inherited Differences within Mitochondrial Complex I Control Murine Healthspan

**DOI:** 10.3390/genes10070532

**Published:** 2019-07-13

**Authors:** Misa Hirose, Paul Schilf, Kim Zarse, Hauke Busch, Georg Fuellen, Olaf Jöhren, Rüdiger Köhling, Inke R. König, Barbara Richer, Jan Rupp, Markus Schwaninger, Karsten Seeger, Christian Sina, Michael Ristow, Saleh M. Ibrahim

**Affiliations:** 1Luebeck Institute of Experimental Dermatology, University of Luebeck, 23562 Luebeck, Germany; 2Energy Metabolism Laboratory, Institute of Translational Medicine, Swiss Federal Institute of Technology (ETH) Zurich, 8603 Schwerzenbach, Switzerland; 3Group of Systems Biology, Institute of Cardiogenetics and Luebeck Institute of Experimental Dermatology, University of Luebeck, 23562 Luebeck, Germany; 4Institute for Biostatistics and Informatics in Medicine and Ageing Research, Rostock University Medical Center, 18057 Rostock, Germany; 5Center of Brain, Behavior & Metabolism, University of Luebeck, 23562 Luebeck, Germany; 6Oscar Langendorff Institute of Physiology, Rostock University Medical Center, 18057 Rostock, Germany; 7Institute of Medical Biometry and Statistics, University of Luebeck, 23562 Luebeck, Germany; 8Institute of Chemistry and Metabolomics, University of Luebeck, 23562 Luebeck, Germany; 9Department of Infectious Diseases and Microbiology, University of Luebeck, 23562 Luebeck, Germany; 10Institute of Experimental and Clinical Pharmacology and Toxicology, University of Luebeck, 23562 Luebeck, Germany; 11Institute for Nutritional Medicine, University of Luebeck, 23562 Luebeck, Germany

**Keywords:** Conplastic mouse strains, mitochondrially encoded complex I, mitochondrial DNA (mtDNA), lifespan, *mt-Nd2*, healthspan, glucose metabolism, mice

## Abstract

Mitochondrial complex I—the largest enzyme complex of the mitochondrial oxidative phosphorylation machinery—has been proposed to contribute to a variety of age-related pathological alterations as well as longevity. The enzyme complex-consisting proteins are encoded by both nuclear (nDNA) and mitochondrial DNA (mtDNA). While some association studies of mtDNA encoded complex I genes and lifespan in humans have been reported, experimental evidence and the functional consequence of such variants is limited to studies using invertebrate models. Here, we present experimental evidence that a homoplasmic mutation in the mitochondrially encoded complex I gene *mt-Nd2* modulates lifespan by altering cellular tryptophan levels and, consequently, ageing-related pathways in mice. A conplastic mouse strain carrying a mutation at m.4738C > A in *mt-Nd2* lived slightly, but significantly, shorter than the controls did. The same mutation led to a higher susceptibility to glucose intolerance induced by high-fat diet feeding. These phenotypes were not observed in mice carrying a mutation in another mtDNA encoded complex I gene, *mt-Nd5*, suggesting the functional relevance of particular mutations in complex I to ageing and age-related diseases.

## 1. Introduction

Mitochondrial DNA encode genes for two ribosomal RNAs and the 22 transfer RNAs, as well as the 13 proteins of mitochondrial respiratory process, i.e., oxidative phosphorylation (OXPHOS) complexes [1,2]. Of the five enzyme complexes of the mitochondrial respiratory process, OXPHOS complex I (NADH:ubiquinone oxidoreductase) is the largest enzyme, and is crucial for cellular metabolism by oxidising NADH (nicotinamide adenine dinucleotide, reduced) and regenerating the NAD^+^ (nicotinamide adenine dinucleotide, oxidised) pool in the mitochondrial matrix [3]. A deficiency in complex I, caused by either the nuclear encoded genome or mitochondrial genome (mtDNA), is reportedly associated with mitochondrial disorders both in children and adults, including Leigh syndrome, cardiomyopathy and encephalomyopathies [4,5].

In addition, the link between mutations in complex I and ageing has been shown in a number of studies using different model systems. Mutations in genes encoding subunits of complex I reportedly increase reactive oxygen species (ROS) levels and extend lifespan in worms and flies by independent mechanisms [6,7]. *Nuo-6* (qm200) worms, carrying a mutation in a conserved subunit of complex I, exhibit an extended lifespan by reducing oxygen consumption and decreasing complex I activities [6]. Flies with a knocked down expression of NDUFS1 (NADH: ubiquinone oxidoreductase core subunit S1), a component of complex I, exhibited a longer lifespan via an increased mitochondrial unfolding protein response and the repression of insulin signalling [7]. Other studies showed that the stability of complex I is a critical factor for the extension of lifespan of mice [8] and worms [9]. In these two studies, opposite theories are suggested. The former indicates that the instability of complex I results in reduced ROS production, which extends the lifespan in mice [8], while the latter shows that the destabilisation of the complex leads to a shortened lifespan in worms [9]. In addition, a recent study demonstrated that increased ROS production, specifically from complex I reverse electron transport, extends lifespan in flies [10].

Apart from studies of nuclear genome encoded complex I genes, mtDNA encoded complex I genes are also reportedly involved in ageing, and this is supported by several studies exhibiting an association between mitochondrial DNA polymorphisms and lifespan in different ethnic populations. These include the A variation at m.5178 in the ND2 gene (*MT-ND2*) accumulating in Japanese centenarians [11], and the A variation at m.9055 in the ATP6 gene (*MT-ATP6*) in French Caucasian centenarians [12]. More recently, a gene-wise analysis of mtDNA in a Turkish population over 90 years of age revealed that nonsynonymous mutations in complex I genes were enriched [13]. Interestingly, most of these variants exhibit protective effects against ageing and age-related diseases, such as reduced lipid levels (anti-atherogenic) [14] and protection from Parkinson’s disease [15]. In addition, variations in mtDNA encoded complex I genes are reportedly associated with age-related diseases, such as metabolic diseases (e.g., type 2 diabetes [16]) and neurodegenerative disorders (e.g., Alzheimer’s disease [17] and Parkinson’s disease [15]), in humans.

Previous studies using model organisms reveal that mutations in mtDNA encoded complex I genes controlled lifespan in worms [18] and fruit flies [19], while no studies have been conducted in mammalian models to date.

Therefore, we evaluated the impact of mtDNA encoded complex I genes on ageing in mice using conplastic mouse strains carrying different single point homoplasmic mutations in complex I genes on the same nuclear background of *C57BL/6J*, i.e., *C57BL/6J-mt^ALR/LtJ^* (B6-mt^ALR^) and *C57BL/6J-mt^BPL/1J^* (B6-mt^BPL^) [20]. The former carries a homoplasmic mutation in mtDNA encoded NADH dehydrogenase subunit 2 gene (*mt-Nd2*), and the latter has a homoplasmic mutation in the mtDNA encoded NADH dehydrogenase subunit 5 gene (*mt-Nd5*). We specifically selected these conplastic strains to compare because they differ only in complex I genes (Appendix A).

## 2. Materials and Methods

### 2.1. Mice and Husbandry

Conplastic mouse strains *C57BL/6J-mt^ALR/LtJ^* and *C57BL/6J-mt^BPL/1J^* were previously generated [20]. *C57BL/6J-mt^ALR/LtJ^* and *C57BL/6J-mt^BPL/1J^* with a backcross of 13 to 18 (*C57BL/6J-mt^ALR/LtJ^*) and 17 to 19 (*C57BL/6J-mt^BPL/1J^*) generations were used for the survival study. To genotype nuclear genome of both *C57BL/6J-mt^ALR/LtJ^* and *C57BL/6J-mt^BPL/1J^* we used the MegaMUGA Mouse Universal Genotyping Array (77,808 SNPs) as described in Appendix A, and greater than 99.9% of SNPs were identical to those of *C57BL/6J* (Appendix A).

Mice had *ad libitum* access to filtered water and autoclaved pellet diet (Altromin, Eastern-Westphalia/Lippe, Germany). The animal facility was maintained at 21 °C on a 12 h light–12 h dark cycle. Mice were allocated into two study groups: longitudinal study group to evaluate lifespan, and cross-sectional study group to evaluate the mice at different ages.

For the high fat diet feeding experiment, eight female mice of *C57BL/6J-mt^ALR/LtJ^* and *C57BL/6J-mt^BPL/1J^* at four weeks of age were fed a high fat diet (EF D12492 (I) 60 kJ% fat, Ssniff, Soest, Germany), while three age- and sex-matched mice of each strain were fed a control diet (EF D12450B mod. LS 13 kJ% fat, Ssniff) over eight weeks.

Animal use and all protocols used in this study were approved by local authorities of the Animal Care and Use Committee (V242-7224. 122-5, and 5-1/16, Kiel, Germany) and performed in accordance with the relevant guidelines and regulations by certified personnel.

### 2.2. Lifespan Study and Determining Age at Death

Eighty-two female *C57BL/6J-mt^ALR/LtJ^*, 82 female *C57BL/6J-mt^BPL/1J^*, 99 male *C57BL/6J-mt^ALR/LtJ^* and 94 male *C57BL/6J-mt^BPL/1J^* were used to evaluate their lifespan, assuring a statistical power to detect a 10% difference in lifespan at a significance level of 0.05 and a power of 0.8 using G*Power [21].

### 2.3. Female Sexual Maturity

Vaginal patency was evaluated as previously described [22].

### 2.4. Plasma IGF-1 Levels

Plasma IGF-1 levels were measured by a commercially available ELISA kit (Mouse/Rat IGF-1 Quantikine ELISA Kit, R&S Systems, Wiesbaden-Nordenstadt, Germany), according to the manufacturer’s protocol.

### 2.5. Statistical Analysis

Survival curves were estimated using the Kaplan–Meier method, and median lifespans with 95% confidence intervals were calculated. The differences of longevity between the strains were analysed using the log-rank method, as sensitivity analysis, Peto and Peto modification of the Gehan test. The R package survival in R version 3.5 was utilized for this [23]. Assuming that a most pronounced effect is visible in female mice and our previous studies demonstrating the impact of mtDNA variants on lifespan is more prominent in females [24], the difference between strains in females was tested first. In a hierarchical way, difference between strains in males and in the sex-combined samples was tested for significance only in case that the previous test was significant at a significance level of 0.05.

Differences in glucose between strains and across time points were compared using a nonparametric analysis of variance for repeated in measures using the package nparLD in R, version 3.5. The effect of strain and age on IGF-1 was investigated in a linear regression model.

Statistical analyses for other functional studies were performed using GraphPad Prism (GraphPad Software, San Diego, CA, USA), and statistical tests used for analysis are indicated in the figure legends.

Statistical tests were performed only for descriptive purposes and descriptive *p*-values are reported.

## 3. Results

### 3.1. A Mutation in the mt-Nd2 Gene Results in a Shorter Lifespan in Mice

First, to evaluate the impact of mutations in complex I genes on lifespan in mice, we performed a longevity study using a large cohort of the two conplastic mouse strains carrying a mutation in *mt-Nd2* (B6-mt^ALR^) and *mt-Nd5* and (B6-mt^BPL^). The female B6-mt^ALR^ mice lived approximately 60 days shorter than the B6-mt^BPL^ mice did (median survival: 829 days in B6-mt^BPL^ and 772 days in B6-mt^ALR^; *p* = 0.0400, log-rank test; Figure 1A and Appendix A), while this significant difference was not observed in males or a sex-mixed analysis (*p* = 0.4923, and *p* = 0.1532, respectively; Figure 1B,C). In both strains, the incidence of spontaneous age-related diseases (i.e., tumours, ulcerative dermatitis and arthritis, Appendix A) and the ageing score (Appendix A) were comparable. The days of vaginal patency and plasma IGF-1 levels in differently aged females showed no difference between the strains (Appendix A).

### 3.2. The mt-Nd2 Mutant Mice Exhibited Mitochondrial Functional Differences under Stress Conditions

Next, we investigated the functional consequence of the *mt-Nd2* variant in the liver mitochondria obtained from the two conplastic mouse strains. Because a significant effect on lifespan was observed only in female mice, only female mice were used in the mitochondrial functional study exclusively. The mitochondrial OXPHOS complex activities values were normalised to the individual values of the citrate synthase (CS) activities. No difference was observed in the levels of the OXPHOS complex enzyme activities (values normalized with CS) between the B6-mt^BPL^ and B6-mt^ALR^ mice in both age groups, except that there was an age-dependent increase in complex III activity levels in both strains (descriptive *p* = 0.0011, young B6-mt^ALR^ vs. aged B6-mt^ALR^; descriptive *p* = 0.0004, young B6-mt^BPL^ vs. aged B6-mt^BPL^; one-way ANOVA, Figure 2A). A Western blot analysis of the liver and heart proteins prepared from the young (3–4 months of age) and aged (18–22 months of age) mice revealed unaltered levels of the OXPHOS complex subunit proteins between the strains in both age groups (Figure 2B,C). The protein levels of three other subunits of mitochondrial complex I protein levels were also evaluated, and no differences were observed between the strains regardless their age group (Figure 2C, right panel).

Primary skin fibroblasts were isolated from the B6-mt^BPL^ and B6-mt^ALR^ mice, and skin fibroblasts cell lines (B6-mt^BPL^ and B6-mt^ALR^) were generated from these primary fibroblasts. Each cell line carried the distinct mtDNA mutations as those in parental mouse strain (Appendix A). A cellular flux analysis of the conplastic fibroblast cell lines demonstrated that the levels of maximal respiration and the spare capacity in B6-mt^ALR^ fibroblasts exhibited a decreasing trend compared to those in the B6-mt^BPL^ fibroblasts (Figure 2D).

Next, primary lymphocytes were isolated from the B6-mt^BPL^ and B6-mt^ALR^ mice, and the mitochondrial superoxide levels were measured by flow cytometry using MitoSOX at the basal (i.e., immediately after the preparation) and immunologically activated (i.e., 24 h after culturing with anti-CD3/anti-CD28 antibodies) status. The activated cells from the B6-mt^ALR^ mice produced less mitochondrial superoxide than did those from B6-mt^BPL^ mice (descriptive *p* < 0.0001, two-way ANOVA; Figure 2E, left). The fold change of increase for superoxide production was lower in the B6-mt^ALR^ lymphocytes than that in the B6-mt^BPL^ cells (descriptive *p* = 0.0005, *t*-test; Figure 2E, right). The mitochondrial membrane potential (MMP) was measured using the same batch of primary lymphocytes used for the superoxide assay. The MMP was increased by the activation, while no difference in the levels of MMP was observed between the strains (descriptive *p* = 0.1457, two-way ANOVA; Figure 2F, left). When the fold change of the MMP levels upon the activation was compared, the B6-mt^ALR^ primary lymphocytes were less abundant than were the B6-mt^BPL^ lymphocytes (descriptive *p* = 0.0445, *t*-test; Figure 2F, right).

### 3.3. Higher Levels of Tryptophan Were Observed in the Cells Carrying the mt-Nd2 Mutation

Regeneration of NAD^+^ is one of the critical functions of mitochondrial complex I. To evaluate whether mutations in complex I affect this function, we determined the NAD^+^ and NADH levels in the liver tissues obtained from the B6-mt^BPL^ and B6-mt^ALR^ female mice. No difference was observed in the NAD^+^/NADH ratio between the strains (Figure 3A, Appendix A). To also obtain insights into the biosynthetic pathways leading to NAD^+^, the levels of tryptophan—an essential amino acid that is degraded into NAD^+^ [25]—was determined by NMR in the skin fibroblast conplastic cell lines carrying the *mt-Nd5* mutation or the *mt-Nd2* mutation (Figure 3B). The levels of tryptophan in the cells carrying the *mt-Nd2* mutation were identified to be higher than those in *mt-Nd5* mutant cells (descriptive *p* = 0.0295, *t*-test; Figure 3B,C), suggesting the m.4738C>A mutation may have an impact on tryptophan metabolism, without altering NAD^+^ levels.

### 3.4. Ageing-Related Pathways and Mitochondrial Functional Pathways Are Altered in mt-Nd2 Mutant Mice

Next, we performed RNA-seq on the liver-isolated RNA obtained from the female B6-mt^BPL^ and B6-mt^ALR^ mice at the age of 3 to 4 months to elucidate the pathways involved in the *mt-Nd2*-related phenotypes. Of the 13,967 expressed genes, 35 genes were differentially expressed between the B6-mt^BPL^ and B6-mt^ALR^ mice (*q* < 0.05; Figure 4A, Appendix A). The genes upregulated in the B6-mt^ALR^ mice included *G0s2*, *Leap2*, *Usmg5*, *Chchd1* and ribosomal protein genes (Appendix A). *G0s2*, a G0/G1 switch gene 2, shows an inhibitory capacity for the lipolytic enzyme adipose triglyceride lipase (ATGL) [26], and the deletion of *G0s2* ameliorates high-fat diet induced body weight gain and insulin resistance [27]. *Leap2* (liver-expressed antimicrobial peptide 2) was recently revealed as an endogenous antagonist of the ghrelin receptor, controlling the blood glucose levels depending upon the nutrient status [28], and *Usmg5* (upregulated during skeletal muscle growth 5) is also increased in B6-mt^ALR^ mice. The gene encodes the protein USMG5, which is also called DAPIT (diabetes-associated protein in insulin-sensitive tissues). This protein was initially discovered in insulin-sensitive tissues of the streptozotocin-induced diabetic rats [29], and its expression is reportedly higher in cells with a highly aerobic metabolism [30]. These findings indicate that the mutation in *mt-Nd2* has an influence on lipid and glucose metabolism, as well as ATP production in the respiratory chain. *Chchd1* (coiled-coil-helix-coiled-coil-helix domain-containing 1) is one of the recently identified mitochondrial ribosomal proteins, MRPS37 [31], suggesting that the *mt-Nd2* variant affects mitochondrially encoded protein expression as well.

Pathways upregulated in the B6-mt^ALR^ mice included the OXPHOS, MYC target, fatty acid metabolism, E2F target and mTORC1 signalling pathways (Figure 4B). The oncogene MYC reportedly targets genes involved in mitochondrial biogenesis, i.e., protein import, complex assembly and mitochondrial transcription/translation [32]. Recent studies showed that E2F regulates genes involved in mitochondrial functions via directly binding to their promoter regions (e.g., COX8 and CYB5-M), and through interactions with key regulatory factors of mitochondrial biogenesis, such as NRF1/2 and PGC-1 beta [33]. These pathway analysis data are in line with the phenotypes observed in the B6-mt^ALR^ mice, i.e., shorter lifespan (Figure 1) and altered mitochondrial functions (Figure 2). An additional pathway analysis using a different database also pointed in the same direction, i.e., the upregulation of mitochondrial bioenergetics in B6-mt^ALR^ mice (Appendix A)

### 3.5. Earlier Onset of Glucose Intolerance Is Induced by High-Fat Diet Feeding in B6-mt^ALR^ Mice

Lastly, to investigate whether the lifespan-linked *mt-Nd2* mutation also contributes to other age-related phenotypes, we induced a diet-induced diabetes model in the female B6-mt^ALR^ and B6-mt^BPL^ mice. The high-fat diet (HFD) or control diet (CD) was started when mice were 4 weeks old. The HFD-fed mice acquired more body mass than did those fed the CD, while there was no strain difference in the HFD- nor CD-fed mice (Figure 5A). After 8 weeks of HFD feeding, an intraperitoneal glucose tolerance test (IPGTT) was performed, and the levels of glucose intolerance in the B6-mt^ALR^ mice were higher than those in the B6-mt^BPL^ mice, with a different time course in both strains (main effect of the strains: descriptive *p* = 0.0238, interaction between strains and time: descriptive *p* < 0.0001, two-way nonparametric analysis of variance; Figure 5B).

The levels of glucose and insulin in the morning-fasted serum samples were comparable between the two strains in the HFD-fed (Figure 5C), CD-fed groups (Appendix A), and in random-fed groups (Appendix A). The fructosamine levels were also unaltered between the random-fed B6-mt^BPL^ and B6-mt^ALR^ mice (Appendix A), confirming that the basal glucose levels are similar between the strains. The lipid parameter assays revealed that the levels of total cholesterol in the HFD-fed mice mildly increased (Figure 5D) compared with those in the CD-fed mice (Appendix A) and random-fed mice (Appendix A). Interestingly, the levels of total cholesterol and high-density cholesterol (HDL) were lower in the HFD-fed B6-mt^ALR^ mice than those in the HFD-fed B6-mt^BPL^ mice (descriptive *p* = 0.0057 and 0.0038, respectively, *t*-test; Figure 5D), while the levels of low-density cholesterol (LDL) were unaltered, resulting in a lower ratio of HDL to LDL in the B6-mt^ALR^ mice compared to that in the B6-mt^BPL^ mice (descriptive *p* = 0.0054, *t*-test; Figure 5D). Changes in total cholesterol and the ratio of HDL to LDL were not detected in the CD-fed groups and random-fed groups (Appendix A). Despite the unchanged triglyceride levels between the HFD-fed B6-mt^ALR^ and HFD-fed B6-mt^BPL^ mice, we observed that HFD-fed B6-mt^ALR^ mice exhibited lower levels of free fatty acid than did the HFD-B6-mt^BPL^ mice (descriptive *p* = 0.0115, *t*-test; Figure 5C), but this did not occur in the CD-fed mice (Appendix A), suggesting impaired beta oxidation in the B6-mt^ALR^ mice under metabolic stress.

Cellular flux analysis was assessed in the primary hepatocytes isolated from the mice fed with HFD over 8 weeks, indicating the lower levels of basal respiration (descriptive *p* = 0.0389, *t*-test; Figure 5E) and OXPHOS-linked ATP levels (descriptive *p* = 0.0446, *t*-test; Figure 5F) in the cells prepared from the B6-mt^ALR^ mice than those in the cells from the B6-mt^BPL^ mice.

We also observed a mildly delayed glucose clearance in the CD-fed B6-mt^ALR^ compared to that of the CD-fed B6-mt^BPL^ mice again with a different time course in both strains (main effect of the strains: descriptive *p* = 0.0062, interaction between the strains and time: descriptive *p* = 0.0017, two-way nonparametric analysis of variance; Appendix A). Furthermore, two independent IPGTT experiments, using regular chow-fed B6-mt^BPL^ and B6-mt^ALR^ mice, demonstrated a pattern of glucose clearance that was similar to that of the CD-fed mice (main effect of the strains: descriptive *p* = 0.3062, interaction between the strains and time: descriptive *p* = 0.0056, two-way nonparametric analysis of variance, Appendix A), confirming that the impaired glucose metabolism was caused by the mutation in *mt-Nd2*. In line with this result, an indirect calorimetric cage analysis of the regular-chow-fed B6-mt^BPL^ and B6-mt^ALR^ mice showed lower levels of respiratory exchange ratio (RER) in the B6-mt^ALR^ mice compared with those in the B6-mt^BPL^ mice, indicating the preference of fat to glucose as an energy source in the B6-mt^ALR^ mice (descriptive *p* = 0.0382, *t*-test, Appendix A). We did not observe any difference in energy expenditure, locomotor activity, or food intake levels other than water intake (Appendix A).

## 4. Discussion

In the present study, we presented experimental evidence that variations in mtDNA encoded complex I genes, i.e., *mt-Nd2* and *mt-Nd5*, differentially affected lifespan and metabolic phenotypes in mammals. Our results are in line with previously reported studies using Drosophila carrying a mutation in the ND2 gene (*ND2^del1^*), i.e., shorter lifespan as well as impaired fat storage, apart from Leigh syndrome-like (or spontaneous) neurological dysfunction [19,34]. These studies suggested that ND2 mutations led to the shorter lifespan.

Mitochondrial functional studies using liver-mitochondria, cellular and tissue proteins did not exhibit major differences at the basal levels between the B6-mt^ALR^ and B6-mt^BPL^ mice. In fact, the impact of the *mt-Nd2* variant on the complex I activity was minimal, with only a tendency of reduced activity in young B6-mt^ALR^ mice compared to age-matched B6-mt^BPL^ mice. Other OXPHOS complexes activities (complex III, IV and V; values were normalised to the individual CS activity) were also comparable between the two strains. At the same time, when normalised to complex I activity, the activity levels of these three OXPHOS complexes exhibited a tendency towards higher levels in B6-mt^ALR^ mice compared to those in B6-mt^BPL^ mice (CIII/CI, 3.76± 0.3156 in B6-mt^BPL^ vs. 5.028± 0.4504 in B6-mt^ALR^, descriptive *p* = 0.0396; CIV/CI, 2.652 ± 0.4682 in B6-mt^BPL^ vs. 3.139 ± 0.5759 in B6-mt^ALR^, descriptive *p* = 0.5340; CV/CI, 3.142 ± 0.3037 vs. 4.735 ± 0.5585, descriptive *p* = 0.0265; unpaired *t*-test), suggesting that these OXPHOS complexes increase their activities to compensate the mildly reduced complex I activity caused by the *mt-Nd2* variant. This observation may explain why B6-mt^ALR^ mice showed a mild increase of superoxide levels and slightly higher levels of mitochondrial membrane potential at the basal levels, but no large statistically significant differences compared to B6-mt^BPL^ mice. However, once the cells received additional stimuli, i.e., immunological- (for lymphocytes) or metabolic stress (for hepatocytes), the levels of oxygen consumption, OXPHOS-linked ATP production and mitochondrial superoxide were significantly lower in the B6-mt^ALR^ mice than those in the B6-mt^BPL^ mice. This effect is likely caused by the lower levels of spare capacity and maximal respiration in the B6-mt^ALR^ mice. In addition, glucose and lipid metabolism were skewed in this mouse strain. These alterations potentially resulted in a shorter lifespan and a higher susceptibility to diet-induced glucose intolerance in the B6-mt^ALR^ mice. Interestingly the RNA-seq data pointed that the pathways involved in the *mt-Nd2* mutant mice were mitochondrial pathways, e.g., OXPHOS pathways and metabolic pathways as well as the ageing-related mammalian target of rapamycin (mTOR) pathway.

Another potential functional consequence of the *mt-Nd2* variant in the B6-mt^ALR^ mice was the altered levels of tryptophan, but not NAD^+^ levels. Tryptophan is an essential amino acid that is metabolised by the kynurenine pathway through a series of metabolic reactions, and consequently NAD^+^ is synthesised [35]. Tryptophan is only obtained through dietary intake, and no difference was observed in the levels of food intake between the strains in our indirect calorimetric cage data. This result suggests that the higher levels of tryptophan were due to the intrinsic metabolic alteration in the B6-mt^ALR^ mice. A previous study demonstrates that tryptophan induces the phosphorylation of the mTOR, and accelerates non-alcoholic fatty liver disease in mice [36], suggesting that the higher levels of tryptophan may contribute to shorten the lifespan in the B6-mt^ALR^ mice by activating the mTOR pathway, which is in line with our RNA-seq data.

Interestingly, the m.4738C>A mutation, which is carried by the B6-mt^ALR^ mice, causes the amino acid substitution of leucine to methionine at the 276th peptide (Leu276Met). The 276th peptide of human ND2 is also leucine, and this residue is known as the binding site for 1,2-dioleoyl-sn-glycero-3-phosphoethanolamine (https://www.ebi.ac.uk/pdbe/entry/pdb/5xtc/protein/17), suggesting that the mutation could potentially affect the binding capacity and/or affinity to glycerolipids, which are the components of the mitochondrial inner membrane. It is tempting to speculate that the *mt-Nd2* mutation alters the structure of mitochondrial complex I, or supercomplex, which may change its function, particularly under stress conditions, without influencing the protein levels of complex I subunits. A study to confirm the structural alteration by the *mt-Nd2* mutation (m.4738C>A), e.g., complex I assembly analysis, will be performed in the future. We also evaluated the potential impact of m.11902T>C variant in B6-mt^BPL^ mice on the protein structure using the same database, however, the amino acid position affected by the variant (Phe54Leu) has not been reported as a potential binding site for any proteins.

In humans, variants in the *MT-ND2* gene, particularly m.5178C>A, exhibit a link with longevity [11] as well as lipid metabolism [14] and the incidence of age-related diseases, e.g., Parkinson’s disease [37]. For this particular variant, the A allele is associated with extreme longevity and is protective for the abovementioned diseases, suggesting that lifespan-associated mtDNA variants are also responsible for age-related disease susceptibility. This effect was particularly observed in female mice. This is consistent with the above-discussed previous report of Drosophila carrying the ND2 mutation, i.e., only female *ND2^del1^* mutant flies exhibited the behavioural phenotype [19]. However, we cannot exclude the possibility that there may be other phenotypes we did not investigate in this study, which this specific variant may affect only in males. For example, we recently reported that mice harbouring another maternally inherited natural mtDNA variant, m.15124A>G in the mitochondrially encoded cytochrome b gene (*mt-Cytb*) in complex III, developed spontaneous middle-aged obesity only in males [38]. In the present study, we demonstrated that the B6-mt^ALR^ mice, carrying the *mt-Nd2* (m.4738C>A) variant, showed a slightly but significantly shorter lifespan and a higher susceptibility to diet-induced glucose intolerance than did the B6-mt^BPL^ mice, which carry the wild type C allele at m.4378 in *mt-Nd2* and the mutant C allele at m.11902 in *mt-Nd5*. While the effect of each SNP appears to be variable in different species, it is clear that variations in the ND2 genes are linked with ageing and metabolic conditions both in mice and humans.

## Figures and Tables

**Figure 1 genes-10-00532-f001:**
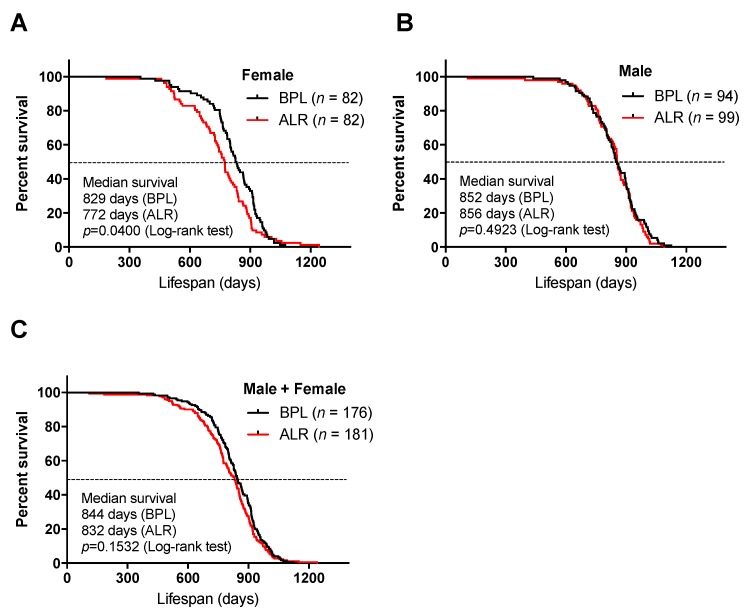
Lifespan study of B6-mt^ALR^ and B6-mt^BPL^ mice. (**A**–**C**) Survival curve of B6-mt^ALR^ and B6-mt^BPL^ mice. Females (**A**), males (**B**) and both sexes (**C**). BPL: B6-mt^BPL^; ALR: B6-mt^ALR^.

**Figure 2 genes-10-00532-f002:**
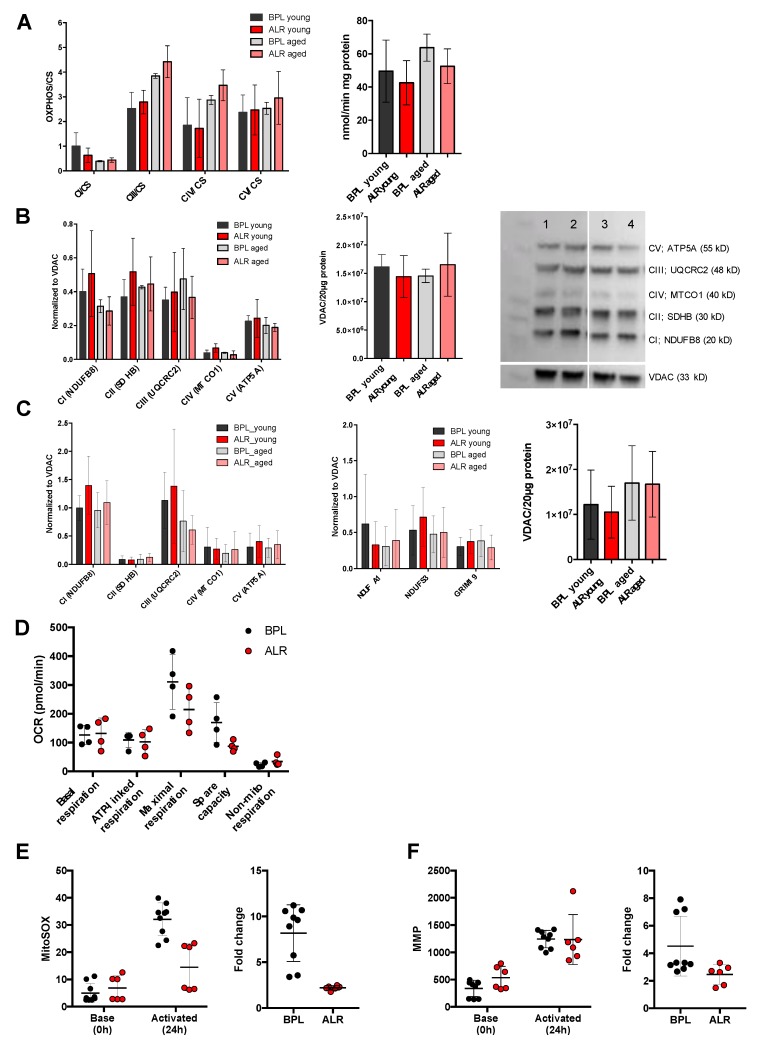
Mitochondrial functional consequence of the mutation in *mt-Nd2* in B6-mt^ALR^ female mice. (**A**) Oxidative phosphorylation (OXPHOS) complex activities were measured in liver mitochondria isolated from B6-mt^BPL^ and B6-mt^ALR^ mice. OXPHOS activities were normalized to each citrate synthase (CS) activities (left panel). No significant difference in activities was detected between the strains in both age groups. Activities in complex III normalised to CS activities were increased in ageing in both strains, however, no difference was observed between strains. CS activities are shown in the right panel. BPL: B6-mt^BPL^; ALR; B6-mt^ALR^; young: 3–4-month-old mice; old: 18–22 months old. *n* = 15 (BPL, young), *n* = 14 (ALR, young), *n* = 4 (BPL, aged), and *n* = 3 (ALR, aged). Descriptive *p* = 0.0011 (BPL, young vs. BPL, aged), descriptive *p* = 0.0004 (ALR, young vs. ALR, aged), one-way ANOVA. (**B**) Quantified values of Western blotting of liver samples demonstrated comparable protein levels of mitochondrial OXPHOS subunits (left panel). *n* = 3/strain (young), *n* = 4/strain (aged). Same samples tested in the left panel were quantified for VDAC (middle panel). Representative Western blotting picture of liver samples (right panel). 1: young B6-mt^BPL^; 2: young B6-mt^ALR^; 3: aged B6-mt^BPL^; 4: aged B6-mt^ALR^. (**C**) Quantified values of Western blotting of heart samples showed comparable protein levels of mitochondrial OXPHOS subunits (left panel). *n* = 6 in each group. Levels of other complex I subunit proteins, i.e., NDUFA1, NDUFS3 and GRIM19, were unaltered in both strains. Same samples tested in the left panel were evaluated (middle panel). Same samples tested in the left and middle panels were quantified for VDAC (right panel). (**D**) Oxygen consumption was evaluated in skin fibroblast cell lines generated from B6-mt^BPL^ and B6-mt^ALR^. The levels of maximal respiration and spare capacity showed a trend of less in B6-mt^ALR^ fibroblasts compared with B6-mt^BPL^ cells. (**E**) Mitochondrial superoxide was measured in primary lymphocytes immediately after the isolation (base, 0h) and 24h-activation with anti-CD3 and anti-CD28 antibodies. Activated lymphocytes from B6-mt^ALR^ produced significantly less mitochondrial superoxide than those from B6-mt^BPL^. The fold change of the MitoSOX levels by activation was significantly lower in B6-mt^ALR^ cells than B6-mt^BPL^ cells. Values from viable cell population that were negative for Annexin V were taken for the analysis. Descriptive *p* < 0.0001, two-way ANOVA (left). Descriptive *p* = 0.0005, *t*-test (right). *n* = 9 (B6-mt^BPL^), *n* = 6 (B6-mt^ALR^). (**F**) Mitochondrial membrane potential was evaluated in the same primary lymphocyte samples as panel E. The geometric means of TMRE in MitoTrackerGreen positive viable cell population was taken as the mitochondrial membrane potential (MMP) value. The fold change of MMP by activation was significantly less in B6-mt^ALR^ lymphocytes than B6-mt^BPL^ cells. descriptive *p* = 0.0445, *t*-test.

**Figure 3 genes-10-00532-f003:**
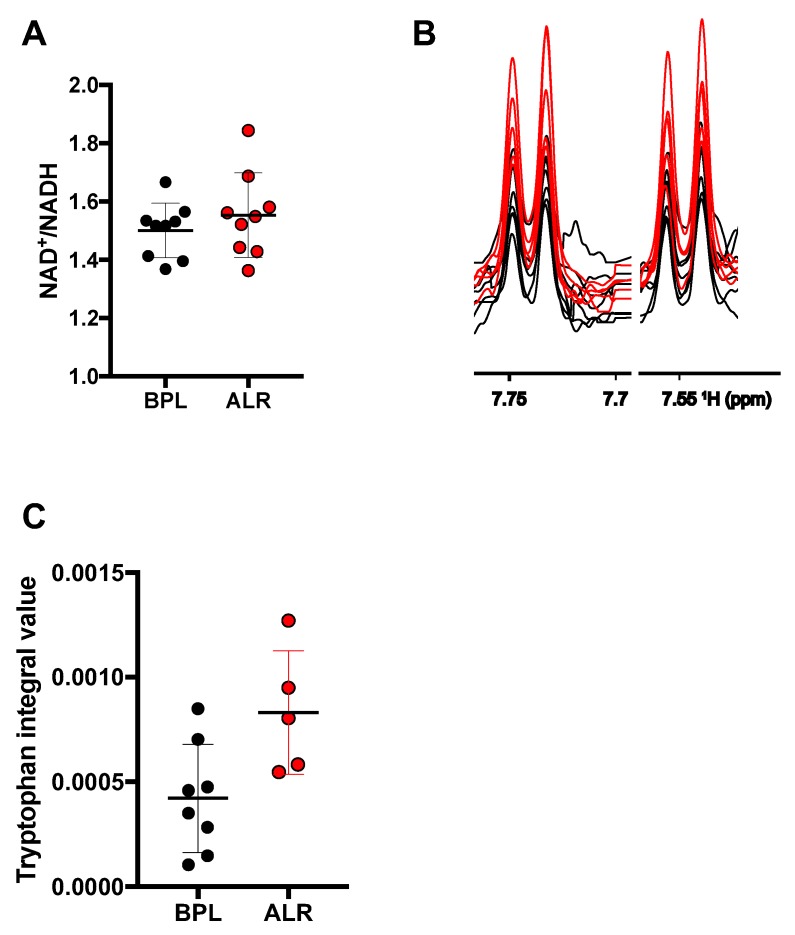
The mutation in *mt-Nd2* is associated with alteration of tryptophan levels. (**A**) A ratio of NAD^+^ to NADH was measured in liver tissues of young (3 months old) female B6-mt^BPL^ and B6-mt^ALR^ mice. No difference was observed (descriptive *p* = 0.3737, *t*-test). *n* = 9/strain. BPL: B6-mt^BPL^; ALR: B6-mt^ALR^. (**B**) A region of the NMR spectra of a well-separated signal of tryptophan in B6-mt^ALR^ fibroblast cell line (red) and B6-mt^BPL^ fibroblast cell line (black) is displayed. *n* = 5 (B6-mt^ALR^ cell lines), *n* = 7 (B6-mt^BPL^ cell lines). (**C**) Tryptophan levels identified in panel B were quantified and compared between the groups. The mean value of the two signals was compared. Descriptive *p* = 0.0295, *t*-test.

**Figure 4 genes-10-00532-f004:**
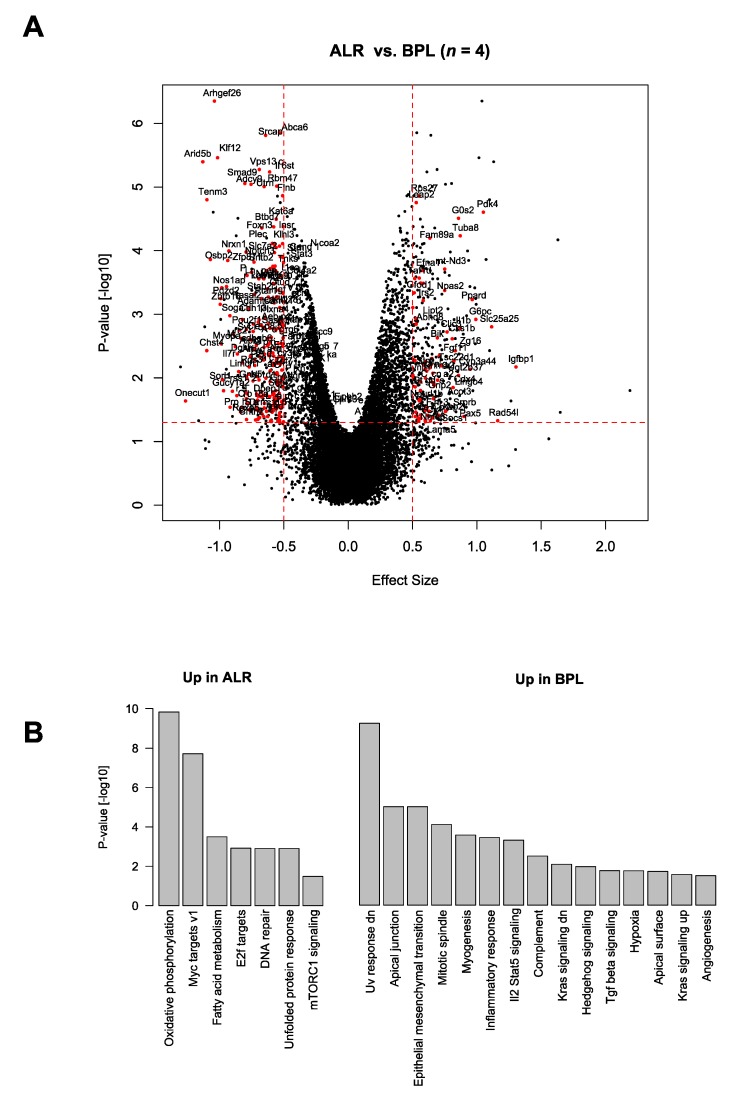
Transcriptome analysis revealed mitochondrial and age-related pathways were affected by the *mt-Nd2* mutation. (**A**) Differentially expressed genes between B6-mt^ALR^ and B6-mt^BPL^ mice. The volcano plot shows the effect size versus the -log10 *p*-value of differentially regulated genes. Genes with the expression levels *p* < 0.01 were plotted in red. *n* = 4/strain, female. BPL: B6-mt^BPL^; ALR: B6-mt^ALR^. (**B**) Hallmark pathway analysis show the pathways upregulated in B6-mt^ALR^ mice (left) and upregulated in B6-mt^BPL^ mice (right).

**Figure 5 genes-10-00532-f005:**
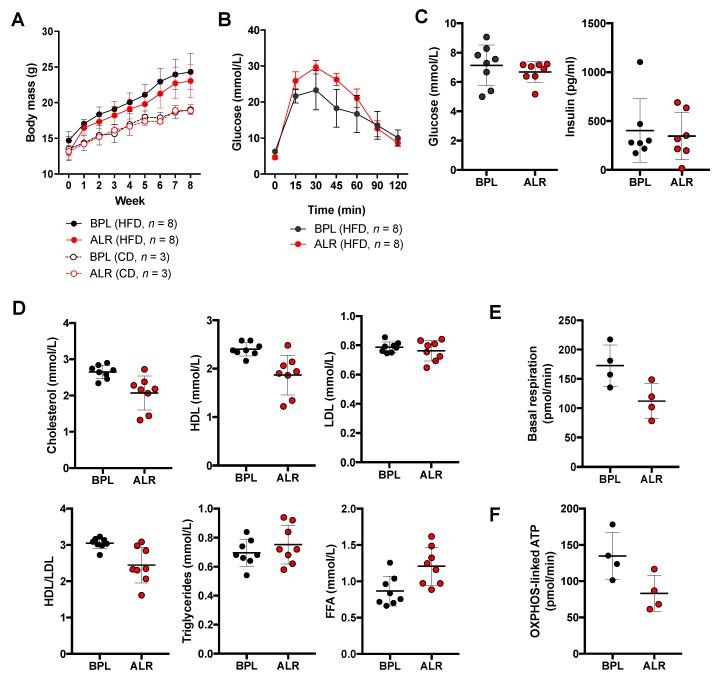
The B6-mt^ALR^ mice exhibited higher susceptibility to diet-induced glucose intolerance. (**A**). Female mice were fed with high fat diet (60 kJ% fat, *n* = 8/strain) or control diet (13 kJ% fat, *n* = 3/strain) for 8 weeks starting at 4 weeks of age. Body mass and weight gain due to the high fat diet were unaltered between strains. HFD: high fat diet; CD: control diet. BPL: B6-mt^BPL^; ALR: B6-mt^ALR^. (**B**) Intraperitoneal glucose tolerance test was conducted after 8 weeks of feeding. B6-mt^ALR^ mice exhibited impaired glucose tolerance compared to B6-mt^BPL^. Main effect of strains: descriptive *p* = 0.0238, interaction between strains and time: descriptive *p* < 0.0001, two-way nonparametric analysis of variance. (**C**) Levels of glucose and insulin were determined in morning-fasted serum samples obtained HFD-fed B6-mt^BPL^ and B6-mt^ALR^ mice. (**D**) Lipid profiles were measured in the same samples as tested in **c**. Cholesterol, descriptive *p* = 0.0057; HDL, descriptive *p* = 0.0038; HDL/LDL, descriptive *p* = 0.0054; FFA, descriptive *p* = 0.0115; *t* test. HDL: high-density lipoprotein; LDL: low-density lipoprotein; FFA: free fatty acids. (**E**,**F**) Primary hepatocyte prepared from mice fed with HFD demonstrated the levels of basal respiration and OXPHSO-linked ATP production were lower in hepatocytes from B6-mt^ALR^ than those from B6-mt^BPL^. Descriptive *p* = 0.0389 (**E**), descriptive *p* = 0.0446 (**F**), *t*-test.

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
