# Peer review of "Maternally Inherited Differences within Mitochondrial Complex I Control Murine Healthspan"

_genes, 2019, doi:10.3390/genes10070532_

Round 1

Reviewer 1 Report

The revised manuscript addressed my comments well.

Author Response

Dear Reviewer 1,

Thank you very much for the opportunity to revise our manuscript.

Please find attached a pont-by-point reply to your valuable comments and the revised version of the manuscript.

We look forward to your revaluation of our work.

Sincerely yours,

Saleh Ibrahim

Reviewer 2 Report

Hirose et al. show in their paper "Maternally inherited differences within 2 mitochondrial Complex I control murine health span" data on two conplastic mouse strains that show very little difference in their mtDNA sequences. Nevertheless, in one of these strains a difference in the lifespan of female (but not in male) mice was observed. Additional data is presented to link this outcome to a mutation in the  mt-Nd2 coding region.

The paper is well written and easy to understand. The differences between the strains are relatively small but statistically significant. This is not unexpected, as differences between mtDNA haplotypes seem to be relatively small and better observable under challenging conditions, as Hirose et al. show with higher susceptibility of said mouse strain to glucose intolerance induced by high-fat diet feeding.

So the differences are small and only observable in females, I think that analysing mtDNA genetically closely related conplastic mouse strains is definitely an important way to finally elucidate some of the functional consequences of single point mutation in mtDNA. Therefore this paper is very interesting for the community.

Minor points

I would like to see the survival times of the (female) mt-Nd5 mice in a figure as in the main text.

Line 119: Eighty two female C57BL/6J-mtALR/LtJ, 82 female C57BL/6J-mtBPL/1J, male 99 C57BL/6J-mtALR/LtJ, and… Should read "99 male"

Author Response

Dear Reviewer 2,

Thank you very much for the opportunity to revise our manuscript.

Please find attached both a point-by-point reply to your comments and the revised version of the manuscript.

We look forward to your re-evaluation of our work.

Sincerely yours,

Saleh Ibrahim

Reviewer 3 Report

This well-written manuscript by Hirose et al describes a variety of metabolic effects of a pair of conplastic mous strains carrying homoplasmic mutations in relevant mitochondrial DNA-encoded RCI genes. While this study is mostly descriptive in nature, it merits high originality as it meets an important need in the field: assessment of RCI mutations in the context of metabolism and phenotypes associated with age-related diseases in a mammalian model. 

In general, the experiments are quite standard, adequately described, and appropriately performed and analyzed. Some of the foundational data in Figure 2 (B, C, D) appear to contain more variation (as exemplified by large error bars) than is typical for these assays and therefore may mask some real biological differences. It would be in the authors' interest to clean these data up by additional Western blot analysis. However, while this suggestion would slightly improve the interpretability of this study, this reviewer will leave it up to the authors as to whether or not they wish to pursue this suggestion. 

An additional suggestion is to moderate some of the claims of significance. This authors are to be commended in the choice and execution of their statistical analyses. However, in this reviewer's view, the authors place too much emphasis on statistical relevance and not enough emphasis on potential biological relevance. In particular, while there is a statistical difference in percent survival in female mice strains, the difference is slight. The authors use this slight statistical difference in one sex as the basis to label the ALR strain as "short lived" later in the manuscript, which seems to be a bit of an overreach. Lines 411 through the end of the manuscript really should be moderated as well with respect to the "significantly shorter lifespan" and the last reference to "age-associated diseases", as the data only demonstrate a slight difference in one sex, and moderately different glucose clearance dynamics is likely not enough to claim an age-associated disease. Throughout the rest of the manuscript, the authors exhibit commendably accurate descriptions of the data. 

Author Response

Dear Reviewer 3,

Thank you very much for the opportunity to revise our manuscript.

Please find attached, both a point-by-point reply to your valuable comments and the revised version of our manuscript.

We look forward to your re-evaluation of our work.

Sincerely yours,

Saleh Ibrahim

This manuscript is a resubmission of an earlier submission. The following is a list of the peer review reports and author responses from that submission.

Round 1

Reviewer 1 Report

Hirose and colleagues in this manuscript use unique mouse model and describe differences in the life span of two complastic mouse strains carrying single point mutation in the mitochondrial DNA in either ND2 or ND5 genes, which encode for subunits of the respiratory complex I. Alterations in CI subunits have been described to affect life span in flies and worms but this has not been demonstrated in mammals. Interestingly, authors find that only ND2 mutant females have a significantly shorter life span than the ND5 mutant mice. Numerous analysis were performed to unravel the molecular mechanism behind this difference. Major findings suggest the involvement of tryptophan in metabolic alteration and susceptibility to glucose tolerance when mice are challenged with a high fat diet. Mitochondrial dysfunction (ROS production and altered mitochondrial membrane potential) was observed in activated lymphocytes but no effect in CI or other respiratory complexes was observed in liver. It is unclear why so many different tissue were analyzed and same analysis was not confirmed in another tissue. It is disappointing that authors did not explore nor discussed the sex differences. Although it is appreciated the extensive data in various tissues. There is no confirmation that what is observed in one tissue like liver, it occurs in skin fibroblast or in heart tissue or lymphocytes.  For example, tryptophan levels were analyzed in primary skin fibroblast but this is not confirmed in liver tissue. However, authors discuss that alterations in tryptophan levels might be the culprit of reduced lifespan in the ND2 mice without having analyzed it in any of the tissues (liver, heart, lymphocytes). In summary, this is a very interesting study trying to understand the role of mtDNA mutations in lifespan and metabolic consequences however there are various points of concern in the manuscript.

Comments

1.    It is unclear whether only female mice were used for all biochemical analysis in in figures 2, 3A and 4. Please clarify this and indicate sex used in each figure as in Fig 5 legend.

2.    In figure 1. authors compare the ND2 (ALR) to the ND5 (BPL) but there is not comparison with the C57BL/6j. It would be helpful to show the lifespan of a C57 strain without any mutation in the mtDNA in Fig 1 or refer to it in the text.

3.    In figure 2 it is unclear why liver tissue was used to study the enzymatic activity of respiratory complexes and the steady-state level of their subunits was performed in heart. Why? This occurs in many of the experiments using different tissues but not explanation or confirmation of same findings in other tissues. Authors assume that all tissues are equally affected and this might not be the case. 

4.    It will be nice to include a representative western blot of the respiratory complex subunits from liver.

5.    In figure 2A, line 164-165 authors mention that there is an age dependent difference in CIII activity. Is this due to a decrease in CS activity or to an increase CIII activity. It will be helpful to show if there are variations on CS activity of this remains constant with age. There are reports describing age differences in the activity of OXPHOS complexes.

6.    In Figure 2D, it is Unclear whether the seahorse data was normalized after OCR measurements. Because cells could grow at different rates overnight an effort must be made to normalize data on day of experiment either by cell number (counting cells again) or by total protein per well.

7.    Lines 217-226 and fig 3. Authors show that NAD/NADH does not change in young liver and then that tryptophan levels are increased in skin fibroblast from ND2 mouse. Conclusions in Line 225-226 is that mutation in ND2 “have an impact on tryptophan metabolism without altering NAD levels”. However, NAD/NADH was not measured in fibroblast nor tryptophan was measured in liver. In the same note, authors discuss that tryptophan might be responsible for the changes in life span (lines 350-359). Again, there is an assumption that mutation affects equally all tissues and the conclusion is not backed up by experimental data. If increased tryptophan is responsible for shorter lifespan, then one would expect that the liver in ND2 female will have increased tryptophan levels but not males. Unfortunately, this was not investigated.

8.    The discussion section will benefit by including discussion of the sex difference and why a mutation will affect more females than males. 

9.    Lines 360-369. Authors discuss the ND2 mutation and its possible alteration to glycerolipids. But there is no discussion on how the mutation in the ND5 gene alters any structural or domain in this protein. If the amino acid affected by the mutation in ND5 is not functionally relevant then one will not expect any phenotype. This is not discussed. Mutations in ND5 are frequent cause of mitochondria diseases such as MELAS, MERRF and Leigh syndrome. 

10. Authors should discuss more the function of ND2 and ND5 subunits in CI and how defects in them can affect assembly of CI and how this might have consequences in other mitochondrial parameter such as ROS production or altered mitochondrial membrane potential without affecting CI activity.

Author Response

Dear Reviewer 1,

We would like to thank you for your encouraging comments and advice. 

Please find our point-by-point response to your comments.

Sincerely yours,

Saleh Ibrahim

Reviewer 2 Report

The MS tested the impact of mtDNA-encoded complex I genes on aging using mice. The experiment is well designed and the results are well described. I don't have major concerns on this MS.

Minor comments:

The font of Fig.2 legend is not consistent.

The quality of Fig.3b is low.

The gene names in Fig. 4a are hard to read.

Author Response

Dear Reviewer 2,

We would like to thank you for your encouraging comments and advice.

Please find out point-by-point response to your comments.

Sincerely yours,

Saleh Ibrahim
